# Community-Level Phenotypic Adaptations of Small Mammals Under Rain-Shadow Dynamics in Baima Snow Mountain, Yunnan

**DOI:** 10.3390/ani16010091

**Published:** 2025-12-28

**Authors:** Yongyuan Li, Guangzhi Chen, Mengru Xie, Yihao Fang, Feng Qin, Wenyu Song

**Affiliations:** 1Vector Laboratory, Institute of Pathogens and Vectors, Yunnan Provincial Key Laboratory for Zoonosis Control and Prevention, Dali University, Dali 671000, China; liyongyuan2025@126.com (Y.L.); yanmiechenyuan@163.com (G.C.); xiemengru1107@163.com (M.X.); qinfeng0609@163.com (F.Q.); 2Institute of Eastern-Himalaya Biodiversity Research, Dali University, Dali 671003, China; fangyh2025@126.com

**Keywords:** appendage allometry, community weighted mean, environmental gradient, osmoregulatory capacity, phenotypic response, renal morphology

## Abstract

Community-level functional traits reflect species’ responses to environmental factors and their contributions to ecosystem functions. The contribution of energy can directly reflect how species in an ecosystem utilize resources, interact, and impact the environment in response, which in turn affects the structure and function of ecosystems. This study evaluates the associations between environmental factors and community-aggregated trait values in the Baima Snow Mountain, Yunnan, China, to examine the classic Bergmann’s and Allen’s rules, as well as renal phenotypic variations accounting for the local aridity gradient resulting from the intensive rain-shadow dynamic. A total of 807 small mammal individuals were recorded belonging to four orders, eight families, and 24 species. A dataset of traits corresponding to temperature, productivity, and water availability was compiled. Ordinary least squares (OLS) regressions were employed to determine the associations between community-weighted mean trait values and selected environmental predictors. We performed Mantel tests to assess the strength of the influence of transition of species compositions, which is measured as the Bray–Curtis dissimilarity index, on community-level trait variations. We found that, at the community level, variations in body sizes were consistent with Bergmann’s rule, while variations in appendage allometry violated Allen’s rule but were partly explained by productivity and habitat conditions. Surprisingly, we found that renal morphology relating to osmoregulatory capacity did not align with the expectation of water constraint, but its converse.

## 1. Introduction

The adaptation of phenotypic features to environmental gradients constitutes a part of the most extensively studied and challenging research areas in ecology [1,2]. Bergmann’s rule posits that endotherms possess larger body size in colder regions [3]. Another relevant notion was Allen’s rule, which stated a positive relationship between ambient temperature and extremities in endotherms [4]. Both hypotheses were grounded on the physical principle that reducing body surface area minimizes heat loss [5], and they have continued to attract extensive attention in the centuries following their initial proposal.

While some ecologists have confined their examination of Bergmann’s rule to within-species or genus-level analyzes [6,7,8], others contend that thermoregulatory mechanisms operate consistently across endothermal taxa, thereby permitting interspecific evaluation of Bergmann’s rule [9]. Nevertheless, these predictions do not always hold and appear contingent upon factors including locality [10] and taxonomic group [11,12]. While strong evidence was found to support Bergmann’s rule [13,14], some studies found no evidence or showed the opposite patterns [15,16].

The validity of Allen’s rule, which is an analog to the inconclusive status of Bergmann’s rule, has been both confirmed and refused depending on empirical studies. Most studies found evidence for the reduction in size of extremities following decreasing temperature, which supported Allen’s rule [17,18]. In rodents, a global test of Allen’s rule showed that Allen’s rule is confirmed only for tails, and this association is likely to be driven by adaptation to the cold, rather than warm, temperatures [19]. On the contrary, the South American genus *Ctenomys* does not follow Allen’s rule but its converse, indicated by that tail proportion relative to body mass increasing with latitude [20]. The counterexample of Allen’s rule was also found in a wood mouse population, with considerable reduction in ear and tail lengths following climate warming [21]. These preceding investigations underscore the context-dependent character of thermoregulatory influences on morphological variation and emphasize the necessity for expanded empirical data across diverse geographical regions and taxonomic groups. Moreover, recent advances have emphasized the critical importance of accounting for appendage allometry (size relative to body size) when evaluating Allen’s rule, as neglecting this factor may compromise the interpretation of appendage size patterns [22].

The inherent complexity of mechanisms underlying spatial variations in morphology largely stems from the synergistic interactions among various contributing factors. Recent research has revealed the complementarity between Bergmann’s rule and Allen’s rule in body sizes and bill length of birds [23]. The growing number of studies also provides evidence for several complementary hypotheses attributing morphology variations to resource availability [24] and starvation resistance [25]. The resource rule, describing how the availability and quality of consumed resources determine the energy expenditure requirements in endotherms [26], may provide a complementary explanation to Bergmann’s and Allen’s rules. This rule implies the positive correlation between body size and productivity, which is supported by findings in desert mammals [27] and global rodent species [28].

The osmoregulatory capacity represents another overlooked phenotypic adaptation to aridity gradients [29]. Similar to cold exposure, drought functions as an environmental filter, compelling animals to optimize water utilization strategies [30,31]. Rymer et al. [31] listed nearly 30 characteristics belonging to three categories (i.e., behavior, physiology, and morphology), which promote mammals’ ability to cope with water restriction. Among these, renal structure, especially medullary thickness, which directly points to the maximum length of the Henle loop [31], was frequently used as an estimate of urinary concentrating ability [32,33,34,35]. This correlation suggests that animals inhabiting arid regions exhibit an increased proportion of renal medulla to optimize water utilization, a pattern that has been demonstrated at the community level and is reflected in the clustered functional structure of communities across both local [36] and global scales [37].

Our research seeks to decipher the relative contributions of thermal, productivity, and humidity gradients in driving functional trait assembly of small mammal communities in the rain-shadow-impacted Baima Snow Mountains. We predicted the following: (1) consistent with Bergmann’s rule, body sizes would decrease with increasing temperature [38]; (2) according to Allen’s rule, the allometric scaling of appendages would increase with rising temperature [19]; (3) following the resource rule, both body size and appendage dimensions would increase with productivity [28]; (4) in accordance with the water utilization hypothesis, kidney traits associated with osmoregulatory function would decrease with greater water availability [31]. These efforts may facilitate our understanding of species’ persistence across environmental gradients through phenotypic adaptation strategies under contemporary global change scenarios.

## 2. Materials and Methods

### 2.1. Study Area

Baima Snow Mountain is situated in the northern part of Yunling mountain range and the core of Hengduan Mountains, southwest China, spanning the coordinates 27°47′ N–28°36′ N and 98°57′ E–99°21′ E. Its highest peak, Zhalaqueni, rises to 5430 m above sea level (m a.s.l.), in sharp contrast to the Lancang River valley to the west and the Jinsha River valley to the east, both of which lie at approximately 2000 m a.s.l. This high-mountain–deep-valley topography induces pronounced environmental and habitat gradients along both elevational and longitudinal dimensions. The overall climate in this area is a subtropical monsoon climate, with an annual average temperature of 4.7 °C, extreme high temperatures reaching 24.5 °C, and extreme low temperatures dropping to −13.1 °C [39,40]. The average annual precipitation in the region is 650.5 mm, with a frost-free period lasting 129 days. Based on elevation and slope differences, the terrain is categorized into three types: river valley areas, montane regions, and alpine zones, exhibiting distinct vertical and horizontal climate patterns including temperature and precipitation variations [41,42]. A pronounced rain-shadow effect has developed in this region: the high massif blocks the westerly Indian Ocean monsoon, causing most precipitation to fall at mid to high elevations, while the river valleys remain hot and arid, reducing vegetation cover and primary productivity along the Jinsha and Lancang River valleys on both sides of the Mountain [39,43,44]. These conditions support diverse ecosystems from low to high elevation, including hot-dry-valley shrublands, forests, plateau wetlands, and alpine meadows [42]. This mountain also forms the core area of the Baimaxueshan National Nature Reserve, which is designated under China’s protected area system for in situ conservation of endemic flora and fauna characterized by high habitat integrity. Seasonal drought stands as another defining climatic feature of these dry-hot valleys, resulting in distinct wet–dry seasons at Baima Snow Mountain. Precipitation is primarily concentrated in June to October, with December to April in winter and spring being the driest season [40].

### 2.2. Small Mammal Sampling

To minimize biases arising from seasonal variation in population fluctuations, age structure, and reproductive stages, field investigations were conducted during approximately the same period in three years—June 2017, July 2023, and May–June 2024, corresponding to the early rainy season in the study area. Standardized field sampling following Song et al. [45] was implemented across elevational gradients at 400 m intervals spanning river valleys (2000 m a.s.l.) to alpine screes (4400 m a.s.l.) on both windward (west) and leeward (east) slopes of Baima Snow Mountain. Transect-based surveys involved 1000 trap-nights per transect, utilizing Sherman live traps, snaps, and bucket pitfalls for small mammal inventory. The Sherman live traps were baited with sugar-free oatmeal, and the snaps were baited with fresh peanuts, while the buckets were deployed on the potential run path of small mammals without bait. The traps were deployed on the first day, checked and re-baited the following morning, and relocated to another transect/sub-transect on the third day.

The captured small mammals were numbered and measured for conventional morphological features, including body weight (BW), head–body length (HB), tail length (TL), hind foot length (HF), and ear length (EL) [46]. The kidneys were extracted and fixed in Bouin’s solution for 48 h, then transferred to and preserved in 75% ethanol for further analysis. The captured individuals were first identified in the field based on morphological characteristics, and those that could not be identified in the field were identified with DNA barcoding technology by using the CytB mitochondrial gene segment.

### 2.3. Community-Level Trait Variables

The morphological traits examined in this study included five external phenotypic features and two renal features. For external measurements, we transformed the values of TL, HF, and EL as their ratio to head–body length to account for appendage allometry [20,22].

For renal variables, the kidneys were incised along a mid-sagittal plane that bisected the papillae using a surgical blade. The slices were manually processed, with an average kidney thickness of 5.5 mm and consistent slicing direction. Measurements of length, width, and height of the kidneys, alongside the cortex thickness and medulla thickness (Figure 1), were obtained using the automated digital microscope ZEISS Smartzoom with 0.5×/0.03 FDW78 mm lens: German Carl Zeiss AG (Oberkochen, Germanay) [47]. Following Heisinger and Breitenbach [48], we then calculate the relative medullary thickness (RMT) and percent medullary thickness (PMT) as follows:$$\mathrm{RMT}\;=\;\frac{10\;(\mathrm{absolute}\;\mathrm{medullary}\;\mathrm{thickness})}{\sqrt[{3}]{\mathrm{kidney}\;\mathrm{volume}}}$$$$\mathrm{PMT}=\frac{100\;(\mathrm{absolute}\;\mathrm{medullary}\;\mathrm{thickness})}{\mathrm{absolute}\;\mathrm{thickness}\;(\mathrm{cortex}+\mathrm{medulla})}$$

The phenotypic variables were measured for each small mammal individual, and the means were derived for the species as trait values. These were then used to calculate the community-weighted mean (CWM), which is obtained by weighting measured species-level functional trait values based on species abundance to derive average trait values at the community level. Following Garnier et al. [49], the CWM was calculated as follows:$$\mathrm{CWM}\;=\;\sum_{i=1}^{n}p_{i}\times {trait}_{i}$$
where $p_{i}$ represents the relative abundance of species *i* in the community, ${trait}_{i}$ denotes the trait value of species *i*. The CWM was computed with the *dbFD* function in the *FD* package [50].

### 2.4. Environmental Variables and Model Fitting

The climate data were derived from WorldClim 2.1 (https://worldclim.org) at a 2.5 s resolution based on near current monthly data selected from 19 years (2001–2020). Data extraction was performed in QGIS 3.38.1 [51]. Out of the 19 bioclimate variables, we excluded those indicating temporal temperature variability but retained only those representing differences between hot and cold conditions, which were: Bio1 (annual average temperature), Bio2 (average diurnal temperature range), Bio5 (highest monthly temperature), Bio6 (lowest monthly temperature), Bio8 (wettest quarterly average temperature), Bio9 (driest quarterly average temperature), Bio10 (warmest quarterly average temperature), and Bio11 (coldest quarterly average temperature). Principal component analysis (PCA) was used to identify the primary gradient from cold to hot conditions. The first principal component (PC1), which explained 88.66% of the temperature variance, was therefore used as the temperature variable in subsequent analyzes. Net Primary Productivity (NPP) was derived from China’s 500 m Annual Net Primary Productivity dataset (2000–2024) in the WGS 1984 coordinate system, provided by the Earth Resources Data Cloud Platform (https://www.gis5g.com). The aridity index (AI) was obtained through the Global Aridity Index and Potential Tasseling Database (Global-AI_PET_v3) in a 30 arc second resolution. This index reflects the humidity in a region, particularly the relationship between precipitation and evaporation or potential evapotranspiration, with a higher AI value indicating greater water availability [52].

The environmental variables were *z*-score standardized to improve normality assumptions and to ensure comparable scales. Then, ordinary least squares (OLS) regression models were constructed separately for CWM of each trait to quantify the linear relationships between individual morphological characteristics (BW, HB, TL/HB, HF/HB, EL/HB, RMT, PMT) and environmental variables (temperature, NPP, AI).

### 2.5. Impacts of Changes in Species Composition on Communities’ Trait Patterns

To determine the association between changes in species composition and CWM values, we calculated the communities’ beta-diversity metric as the Bray–Curtis dissimilarity index using the *vegdist* function in the *vegan* package [53]. Prior to correlation analysis, trait matrices were standardized, with missing trait values excluded via the *na.omit* parameter. Euclidean distances were calculated for CWM for the seven functional traits (BW, HB, TL/HB, HF/HB, EL/HB, RMT, PMT). Then, associations between beta-diversity and CWM distances were assessed using Mantel tests with Spearman’s correlation and 9999 permutations. All analyzes were performed in R 4.4.1. [54] environment.

## 3. Results

### 3.1. Overview of Species Composition

During the study period (total trap-days = 14,794), we captured 807 small mammals encompassing four orders, eight families and 24 species (*Apodemus ilex*, *Apodemus latronum*, *Apodemus chevrieri*, *Niviventer confucianus*, *Niviventer andersoni*, *Niviventer excelsior*, *Rattus tanezumi*, *Eothenomys custos*, *Neodon irene*, *Tamiops swinhoei*, *Callosciurus erythraeus*, *Dremomys pernyi*, *Sicista concolor*, *Ochotona thibetana*, *Ochotona gloveri*, *Ochotona macrotis*, *Sorex bedfordiae*, *Episoriculus macrurus*, *Blarinella wardi*, *Sorex cansulus*, *Sorex thibetanus*, *Uropsilus nivatus*, *Scaptonyx affinis*, *Tupaia belangeri*), with an overall capture rate of 5.45%. Dominance hierarchy analysis identified *E. custos* (35.94%), *A. ilex* (16.6%), and *A. latronum* (10.66%) as the three most abundant species, collectively accounting for 63.2% of total captures.

### 3.2. Relations Between Temperature and Body Size and Appendages

The OLS linear regression analyzes revealed that body size (BW and HB) and appendages (TL/HB, HF/HB, EL/HB) decreased with increasing temperature, indicated by negative correlations (estimate slopes < 0) against PC1 of temperature variables (Figure 2; Table 1). The PC1 of temperature variables demonstrated strong and significant explanatory power for BW (adj*R*^2^ = 0.789, *p* < 0.001) and HB (adj*R*^2^ = 0.689, *p* < 0.001). The explanatory power for TL/HB was also strong and significant (adj*R*^2^ = 0.728, *p* < 0.001). The model fit for HF/HB was significant while the explanatory power was moderate (adj*R*^2^ = 0.451, *p* = 0.009). Association between the PC1 of temperature variables and EL/HB was negative but insignificant (adj*R*^2^ = 0.147, *p* = 0.175).

### 3.3. Relations Between Productivity and Body Size and Appendages

The OLS linear regression analyzes showed that the explanatory power of NPP was not significant (*p* > 0.05) for all five external phenotypic variables: for BW, adj*R*^2^ = 0.019, *p* = 0.638; for HB, adj*R*^2^ = 0.028, *p* = 0.566; for TL/HB, adj*R*^2^ = 0.139, *p* = 0.189; for HF/HB, adj*R*^2^ = 0.227, *p* = 0.085; for EL/HB, adj*R*^2^ = 0.152, *p* = 0.168. However, body size (BW and HB) tended to decrease with increasing productivity, whereas appendages showed the opposite patterns, increasing along the productivity gradient. These tendencies were indicated by negative correlations (estimate slopes < 0) for NPP against body size (BW and HB) but positive correlations (estimate slopes > 0) for appendages (TL/HB, HF/HB, EL/HB) (Figure 3; Table 2).

### 3.4. Relations Between Water Availability and Renal Characteristics

OLS linear regression revealed positive associations between AI and both RMT and PMT (estimated slopes > 0), suggesting increasing trends in these renal metrics with greater humidity. The explanatory effects for AI against both RMT (adj*R*^2^ = 0.584, *p* = 0.001) and PMT (adj*R*^2^ = 0.439, *p* = 0.01) were strong and significant (Figure 4; Table 3).

### 3.5. Associations Between Changes in Species Compositions and CWM Variations

The Mantel test results showed strong and positive correlation between the Bray–Curtis dissimilarity index and BW (Mantel *r* = 0.720, *p* < 0.001), HB (Mantel *r* = 0.706, *p* < 0.001), and TL/HB (Mantel *r* = 0.740, *p* < 0.001). HF/HB was positively but very weakly correlated with Bray–Curtis dissimilarity with significant support (Mantel *r* = 0.199, *p* = 0.046). Nevertheless, EL/HB was not correlated with changes in species compositions (Mantel *r* = −0.065, *p* = 0.709). There was no statistically significant correlation detected between RMT (Mantel *r* = 0.165, *p* = 0.084) and Bray–Curtis dissimilarity, while PMT showed a moderately positive correlation with Bray–Curtis dissimilarity, receiving significant statistical support (Mantel *r* = 0.392, *p* = 0.001).

## 4. Discussion

### 4.1. Variations in Body Size and Appendages

Using overall community trait shifts driven by composition of species’ relative abundances—i.e., community-weighted means (CWM)—enables us to assess morphological responses to environmental variation at the community level. Although this research scheme has been tested in some case studies such as birds and arthropods [55,56,57,58], it remains quite rare in mammals. Temporal and spatial dynamics of the relative abundance of small mammal species allow us to assess many aspects of environmental variability [59,60].

Our first set of results shows that body size decreases markedly as temperature rises, implying that small mammal communities in colder environments are dominated by larger-bodied species. This finding is consistent with Bergmann’s rule. Nevertheless, variations in appendages do not follow Allen’s rule but its opposite. This may be attributed to the complementarity in thermoregulation strategies [23,61]—in cold areas, small mammals increase their body size to reduce heat loss from the body surface, while their appendages become moderately elongated to support other requirements such as mobility, feeding, and predator avoidance [62,63,64]. Using elevation as a proxy of temperature gradient, research from adjacent mountains has described an opposite trend: rodents’ body size and tail length both decrease with increasing elevation, thereby violating Bergmann’s rule and providing slight support for Allen’s rule [65]. The key disparity from this previous observation is that our focal region, Baima Snow Mountain, is strongly influenced by an intense rain-shadow. As a result of the rain-shadow–driven regulation of mountain moisture, the region features low-elevation savanna and high-elevation forest [44], rather than the conventional configuration of low-elevation forest and high-elevation grassland. This context suggests that species may exhibit different morphological responses along the environmental gradient via niche partitioning [66]. Thus, despite contrasts with earlier findings, our study likewise supports the view that thermoregulation and niche partitioning jointly shape species’ morphological variation across temperature gradients [65].

We found no strong evidence to support the relationship between productivity and morphological variations. The resource rule concerning animals’ body sizes remains underestimated [26]. Although statistically non-significant, we observed contrasting patterns in the present study: the estimated slopes were negative for body size variables against productivity, while positive slopes were observed in appendages against productivity. In our study sites, forests at mid-to-high elevations represent high-productivity habitats inhabited by taxa exhibiting morphological adaptations for either cursorial or arboreal locomotion, including elongated hind feet and tails as exemplified in murids (rats and mice) and sciurids (squirrels). In contrast, low-productive alpine meadows and scree habitats support communities dominated by short-footed and short-tailed species such as ochotonids (pikas) and arvicolines (voles). Together, our findings, to a certain extent, suggest that body size variation in small mammals is primarily driven by thermoregulation, while habitat exerts a filtering effect on appendage morphologies [65]. This notion also receives support in the observation that small mammals possess bigger tails in the tropics due to arboreal lifestyles in rainforest ecosystems [18].

### 4.2. Variations in Renal Phenotype

Our results did not support the expected associations that renal medullary thickness enlarged with increasing water constraint, but its converse: RMT and PMT increased with greater water availability (higher aridity index values). We propose two plausible explanations that could elucidate these patterns. First, alpine snowmelt waters within the study area form a series of streams that flow through arid valleys at lower elevations, eventually converging into the western Lancang River and the eastern Jinsha River. Consequently, despite the overall arid climate characteristic of the rain-shadow region, these water resources persist in supporting small mammal communities. Most of our specimens in the hot-dry valley were collected near streams, demonstrating that water availability remains a critical factor limiting the distribution of small mammal assemblages. Second, our study area occupies a unique position at the convergence of both the Palearctic–Oriental biogeographic boundary [67,68] and the transition between northern arid and southern humid climatic zones [69]. Within the vicinity of this area, low-elevation communities are primarily characterized by Oriental taxa, whereas high-altitude communities demonstrate a significantly higher representation of Palearctic components [70,71]. Consequently, the observed positive relationship between water availability and renal water recycling capacity likely does not manifest at the local community level but instead reflects adaptive responses of mammal fauna from distinct biogeographic zones to broad-scale precipitation gradients. Combined, our findings suggest that, within the targeted region, microenvironmental conditions and transitional dynamics between ecoregions may exert a greater influence on water resource utilization than the inherent physiological osmoregulatory capacity [72].

Finally, our distance-based analysis reveals that communities’ beta-diversity is strongly associated with variation in CWMs of BW, HB, and TL/HB, indicating these morphological traits exhibit strong structural synergy and are consistent with niche differentiation theory [73]. In contrast, CWMs of HF/HB and EL/HB were moderately or not correlated with species’ compositional changes. These results suggest that these traits may reflect more complex adaptive strategies shaped by species’ diverse lifestyles (e.g., terrestrial, fossorial, or arboreal) and by differing strategies for foraging and predator avoidance. For example, species such as shrews and voles that rely on olfaction and tactus to excavate tunnels tend to have relatively small ears and short feet, whereas ground-dwelling species that rely critically on audition and greater mobility, such as murids and pikas, have larger ears and hind feet. As a result, community patterns of these traits do not necessarily correspond to species’ compositional change when different species possess similar morphological characteristics [45]. Moreover, we observed a moderate positive correlation between PMT and communities’ beta-diversity, whereas variation in RMT showed no significant pattern. This finding partially supports our hypothesis that renal morphology may be influenced by zoogeographical transitions of taxa along the water-availability gradient. However, given the limitations of our study scale and the constraints on the individuals assessed, this notion warrants further investigation.

## 5. Conclusions

Our results corroborate previous research indicating that community-level phenotypic variations in small mammals result from the complex influences, including climate, productivity, habitat characteristics, and both micro- and macro-scale adaptive strategies [27,28,65]. The temperature dependence of body size variation supports Bergmann’s rule, whereas appendage morphologies deviate from Allen’s rule but are partly explained by productivity and habitat conditions. Furthermore, identifying adaptive strategies employed by animals to respond to water dynamics is critically important for conservation governance, given that the rising frequency of extreme climate events—including severe droughts and intense rainfall—has led to an increasing rate of extinctions, distributional and phenological changes, and species’ range shifts in wild animals in the current era [72,73]. The investigation of phenotypic plasticity in drought resilience represents a notably understudied facet of mammalian ecological research.

## Figures and Tables

**Figure 1 animals-16-00091-f001:**
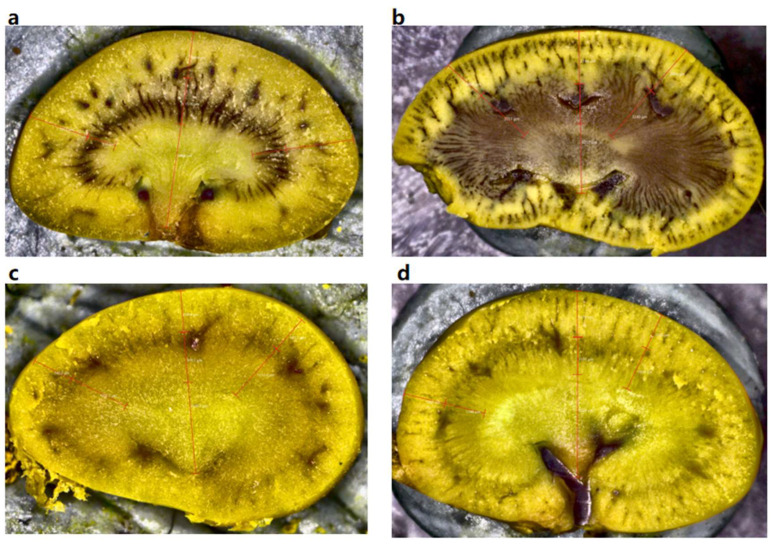
Digital microscopy images of the renal mid-sagittal plane for measuring kidney characteristics for deriving relative medullary thickness (RMT) and percent medullary thickness (PMT). Panels showed examples for (**a**) *Niviventer confucianus*; (**b**) *Ochotona gloveri*; (**c**) *Uropsilus nivatus*; (**d**) *Tupaia belangeri*.

**Figure 2 animals-16-00091-f002:**
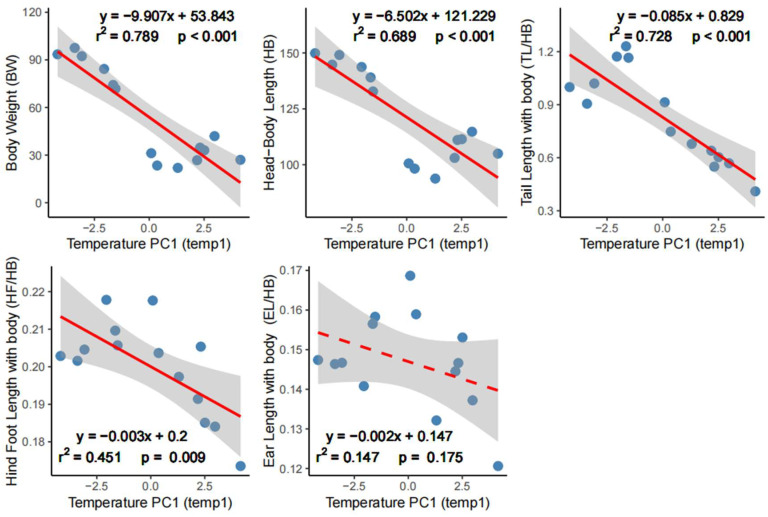
Ordinary least squares linear regressions of the first axis (PC1) of temperature variables against body size and appendage measurements. Appendage measurements, including tail length (TL), hind foot length (HF), and ear length (EL), are standardized as ratios relative to head–body length (HB). Blue dots represent the distribution of community weighted mean (CWM) values of the community; red solid line indicates a significant linear relationship, while the dashed red line denotes a non-significant one; shaded area shows the 95% confidential interval.

**Figure 3 animals-16-00091-f003:**
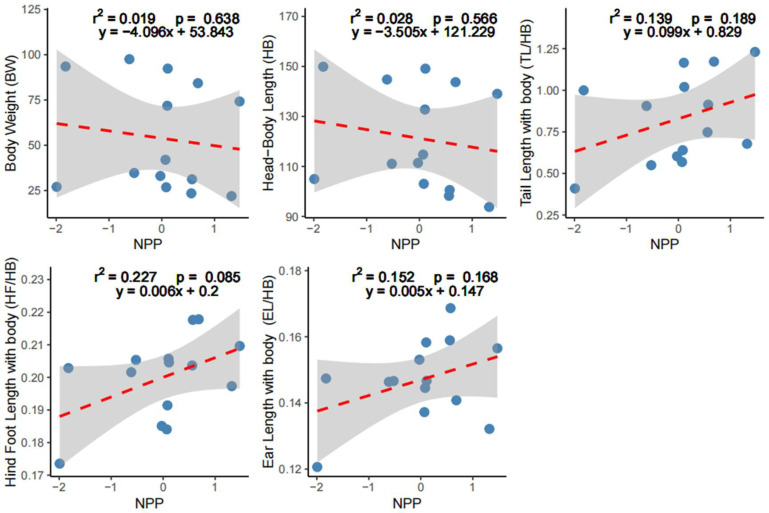
Ordinary least squares linear regressions of net primary productivity (NPP) against body size and appendage measurements. Appendage measurements, including tail length (TL), hind foot length (HF), and ear length (EL), are standardized as ratios relative to head–body length (HB). Blue dots represent the distribution of community weighted mean (CWM) values of the community; the dashed red line denotes a non-significant linear relationship; shaded area shows the 95% confidential interval.

**Figure 4 animals-16-00091-f004:**
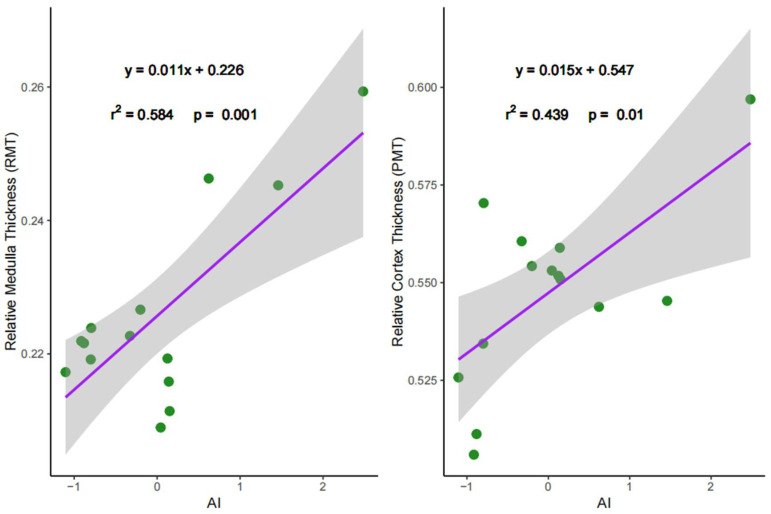
Ordinary least squares (OLS) linear regression for relative medullary thickness (RMT) and percent medullary thickness (PMT) against aridity index (AI). Higher AI value indicates more humid conditions. The green dots represent the distribution of community weighted mean (CWM) values; purple solid line indicates a significant linear relationship, with the shaded area representing the 95% confidence interval.

**Table 1 animals-16-00091-t001:** Ordinary least squares (OLS) linear regression results of PC1 of temperature variables against body weight (BW), head–body length (HB), and ratio to HB of tail length (TL), hind foot length (HF), and ear length (EL).

Phenotypic Variables	Intercept	Estimate	*SE*	*T*	adj*R*^2^
BW	53.843 ***	−9.907 ***	1.480	−6.694	0.789
HB	121.229 ***	−6.502 ***	1.260	−5.162	0.689
TL/HB	0.829 ***	−0.085 ***	0.015	−5.667	0.728
HF/HB	0.200 ***	−0.003 **	0.001	−3.142	0.451
EL/HB	0.147 ***	−0.002	0.001	−1.441	0.147

***: *p* < 0.001; **: *p* < 0.01.

**Table 2 animals-16-00091-t002:** Ordinary least squares (OLS) linear regression results of net primary productivity (NPP) with body weight (BW), head–body length (HB), and ratio to HB of tail length (TL), hind foot length (HF), and ear length (EL).

Phenotypic Variables	Intercept	Estimate	*SE*	*T*	adj*R*^2^
BW	53.843 ***	−4.096	8.494	−0.482	0.019
HB	121.229 ***	−3.505	5.934	−0.591	0.028
TL/HB	0.829 ***	0.099	0.071	1.393	0.139
HF/HB	0.200 ***	0.006	0.003	1.875	0.227
EL/HB	0.147 ***	0.005	0.003	1.468	0.152

***: *p* < 0.001.

**Table 3 animals-16-00091-t003:** Ordinary least squares (OLS) linear regression results of relative medullary thickness (RMT) and percent medullary thickness (PMT) against aridity index (AI).

Phenotypic Variables	Intercept	Estimate	*SE*	*T*	adj*R*^2^
RMT	0.226 ***	0.011 **	0.003	4.108	0.584
PMT	0.547 ***	0.015 **	0.005	3.062	0.439

***: *p* < 0.001; **: *p* < 0.01.

## Data Availability

The experimental data used to support the findings of this study are available from the corresponding author upon request.

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
