# Peer review of "Community-Level Phenotypic Adaptations of Small Mammals Under Rain-Shadow Dynamics in Baima Snow Mountain, Yunnan"

_animals, 2025, doi:10.3390/ani16010091_

Round 1
Reviewer 1 Report
Comments and Suggestions for Authors
The manuscript Community-level phenotypic adaptations of small mammals under rain shadow dynamics examines morphological adaptations of rodents to environmental conditions, in particular, correlation with the well-known Bergman and Allen rules. Despite the fact that these rules have long been formulated, there is still no consensus on the conditions under which they are applicable. The article is based on a lot of factual material, is well illustrated and is of scientific interest. At the same time, when reading, a number of questions arise regarding the methods and material processing.
- Kidneymeasurementprocedure.For renal variables, the kidneys were incised along a mid-sagittal plane that bisected the papillae using a surgical blade, L164-168. Were kidneysectionsperformedmanually,or with a microtome?What was the thickness of the sections? Are the authorssurethat the organ orientationwas the same in allanimalswhenperforming the section?
- The authors discuss morphological adaptations at the community level, but the question arises, what is the ratio between intrapopulation adaptations and changes in community composition? What role do morphological adaptations of each species play, and how great is the impact of a change in species composition?
The authors report, “24 species (overall capture rate: 5.45%). Dominance hierarchy analysis identified Eothenomys custos (35.94%), Apodemus ilex (16.6%) and Apodemus latronum (10.66%) as the three most abundant species, collectively accounting for 63.2% of total captures”, L216-219, this is the total number. But the research was being conducted for 3 years, and there is no information whether there are differences in the abundance and proportion of the species in different years. The description of the regression analysis indicates that a linear regression between body size and temperature is examined, L220-234, and further, the same for humidity. It is unclear from this description whether such factors as species, sex, age, and reproductive state were taken into account.
The body proportions in different species are different (for example, the differences between voles and mice proportions are very significant), so it is necessary to prove that in each season the species ratio was the same, or point out that these changes could be related to changes in the species ratio. In addition, there may be differences in average body size at different phases of the population cycle. The condition of the kidneys can be influenced by age, sex, and participation in reproduction, but the authors do not elaborate on this aspect either. I think the authors took all these factors into account, but for some reason they did not expand on them in the text. I believe that it is necessary to clearly describe the contribution of all these factors or the absence of their impact, if it has not been found.
- It is desirable to get an answer whether such factors as type, gender, age, and reproduction are factored in when analyzing the thickness of the renal layers. If they are not, then perhaps a new analysis will paint a different picture.
- The version of the text that I downloaded contains many technical errors, in particular, a lack of hyphenation. I was puzzled by such words as “charact”, “sligh” “incr” (L43,49,318 and so on) until I realized that these are parts of such words as slightly, characteristics, increased.
- Fig. 1, L181. The figure captions are hardly visible. I think a different font color for the captions (more contrasting, for example, blue) and a larger font size would be better.
In general, this study, even in this form, is of undoubted interest. I think the authors will be able to clarify my questions. If this is not done, similar questions would arise on the part of the readers of the article. The paper is very interesting, but the design of the study needs to be more understandable.
Kind regards
Author Response
Point-by-point response to Comments and Suggestions for Authors
1 Review
Comments 1: The manuscript Community-level phenotypic adaptations of small mammals under rain shadow dynamics examines morphological adaptations of rodents to environmental conditions, in particular, correlation with the well-known Bergman and Allen rules. Despite the fact that these rules have long been formulated, there is still no consensus on the conditions under which they are applicable. The article is based on a lot of factual material, is well illustrated and is of scientific interest. At the same time, when reading, a number of questions arise regarding the methods and material processing.
Response 1: Thank you for your insightful feedback. We sincerely appreciate your recognition of our data richness and scientific value. We have improved the analysis and added related content according to your valuable comments.
Comments 2: In general, this study, even in this form, is of undoubted interest. I think the authors will be able to clarify my questions. If this is not done, similar questions would arise on the part of the readers of the article. The paper is very interesting, but the design of the study needs to be more understandable.
Response 2: We appreciate your positive assessment of the scientific merit of our manuscript and your constructive feedback on the clarity of the research design. We fully recognize that enhancing methodological transparency is essential for reader comprehension. While the current framework effectively captures the dynamics of the rain shadow effect, we acknowledge the need for greater epistemological clarity.
2 General comments
Comments 1: The authors discuss morphological adaptations at the community level, but the question arises, what is the ratio between intrapopulation adaptations and changes in community composition? What role do morphological adaptations of each species play, and how great is the impact of a change in species composition?
Response 1: We greatly value your insightful evaluation. The questions you’ve raised are undoubtedly intriguing and merit exclusive discussion. This study focuses on a rarely explored dimension—phenotypic variations among communities. Consequently, both intraspecies and interspecies levels fall outside the scope of this research. Our additional ANOVA analysis also indicates that the traits used show no significant differences between females and males (see below). These aspects should be addressed in detail in subsequent research efforts.
To deepen our current framework and better address your question concerning the magnitude of effects associated with changes in species composition, we conducted beta-diversity analyses and examined their correlations with the derived CWM values using Mantel tests. Relevant contents have been added to the Methods section (Lines 231-239 ), the Results section (Lines 311-320 ), and the Discussion section (Lines 387-405 ). The results of the new analysis show that the community’s Bray-Curtis dissimilarity index positively and strongly correlate with body size (BW and HB) and one of the appendages (TL/HB), indicating strong structural synergy and consistent with niche differentiation theory. The correlations were moderately positive or non-significant for the rest of the appendages and renal traits. We have discussed these findings in the revision.
Comments 2: The body proportions in different species are different (for example, the differences between voles and mice proportions are very significant), so it is necessary to prove that in each season the species ratio was the same, or point out that these changes could be related to changes in the species ratio. In addition, there may be differences in average body size at different phases of the population cycle. The condition of the kidneys can be influenced by age, sex, and participation in reproduction, but the authors do not elaborate on this aspect either. I think the authors took all these factors into account, but for some reason they did not expand on them in the text. I believe that it is necessary to clearly describe the contribution of all these factors or the absence of their impact, if it has not been found.
Response 2: Although our field investigation spanned three years, surveys were conducted during approximately the same period each year—late May to early July—which corresponds to the early rainy season in the study area. This strategy minimized biases arising from seasonal variation in population fluctuations, age structure, and sex ratios, thereby enabling robust comparisons across transects. We incorporated this justification in Methods section (Lines 155-167 ). On the other hand, our study focuses on transitions in community traits along environmental gradients. Although intraspecific trait variation—such as individual differences related to age and sex—is an important topic, it is beyond the scope of this study.
Comments 3: It is desirable to get an answer whether such factors as type, gender, age, and reproduction are factored in when analyzing the thickness of the renal layers. If they are not, then perhaps a new analysis will paint a different picture.
Response 3: We conducted ANOVA to evaluate the effect of sex on our trait dataset. The results indicated that none of the trait measurements differed significantly between males and females (see the figure below). Therefore, we consider it reasonable not to distinguish between sexes in subsequent analyses of CWM variation among communities. On the other hand, for many of the small mammals we recorded, age was difficult to determine, and, unfortunately, reproductive stages were not documented in the field. We look forward to undertaking additional analysis in future work to address these intriguing questions.

3 Specific comments
Comments 1: Lines 164-168. Kidney measurement procedure. For renal variables, the kidneys were incised along a mid-sagittal plane that bisected the papillae using a surgical blade. Were kidney sections performed manually, or with a microtome? What was the thickness of the sections? Are the authors sure that the organ orientation was the same in all animals when performing the section?
Response 1: Lines 181-185. The slices were manually processed, with an average kidney thickness of 5.5 mm and consistent slicing direction. This protocol was according to previous literature (e.g., Kohli et al. 2019). The small mammals recorded share similar renal anatomy, allowing us to ascertain that sagittal morphology is comparable across species.
Comments 2: Lines 216-219 The authors report, “24 species (overall capture rate: 5.45%). Dominance hierarchy analysis identified Eothenomys custos (35.94%), Apodemus ilex (16.6%) and Apodemus latronum (10.66%) as the three most abundant species, collectively accounting for 63.2% of total captures”, this is the total number. But the research was being conducted for 3 years, and there is no information whether there are differences in the abundance and proportion of the species in different years.
Response 2: We appreciate your interest in potential temporal dynamics. The elevational transects were sampled in the same season spanning three years. Building on previous assumption that species abundance and population structure remain stable in the same seasonal window across different years (e.g., Wen et al. 2014, Biotropica; Liang et al. 2021, Diversity and Distributions), we therefore consider it suitable to combine these data in subsequent analyses. Our dataset also supported this notion—despite interannual fluctuations, the dominant species (Eothenomys custos, Apodemus ilex, and Apodemus latronum) maintain a collective annual abundance above 60%, demonstrating stable biological advantages.Lines155-158
Comments 3: Lines 220-234 The description of the regression analysis indicates that a linear regression between body size and temperature is examined,, and further, the same for humidity. It is unclear from this description whether such factors as species, sex, age, and reproductive state were taken into account.
Response 3: As with the above, we examined the differences in trait measurements between different genders. The ANOVA results showed that the traits used show no significant differences between females and males. Besides, due to age or reproductive states being not recorded, we are not able to test whether there are differences in these traits among individuals.
Comments 4: Lines 43,49,318 The version of the text that I downloaded contains many technical errors, in particular, a lack of hyphenation. I was puzzled by such words as “charact”, “sligh” “incr” until I realized that these are parts of such words as slightly, characteristics, increased.
Response 4: Thank you for your meticulous attention to the technical presentation of our manuscript. We sincerely apologize for the hyphenation errors that caused confusion. This had resulted from an incompatible hyphenation algorithm between our drafting Microsoft Word and the journal's PDF renderer. The Microsoft Word document enabled Western-style line breaks in the middle, which has now been disabled.
Comments 5: Line 181 Fig. 1, The figure captions are hardly visible. I think a different font color for the captions (more contrasting, for example, blue) and a larger font size would be better.
Response 5: Lines 201-204 Fig. 1 Mark as blue and increase the font size from original No.9 to No.10,The caption size has been increased from 9-point to 10-point font. To ensure clearer kidney images, we rephotographed kidneys at various magnification levels.

Reviewer 2 Report
Comments and Suggestions for Authors
Dear Authors,
This manuscript certainly touches on an interesting topic. It is important to study the classic Bergmann's and Allen’s rules in animals in different geographic zones. However, the authors have devoted insufficient attention to describing their results. The manuscript cannot be published in its current form. The title of the manuscript should be amended. The text of the manuscript is jumbled in some chapters and should be moved to the appropriate chapters. In some methodological aspects of small mammal capture, the authors omit important information. This information should be added. The comparative section in the discussion needs to be expanded and additional literature on other geographic zones should be cited. The authors have obtained interesting results, presented them, and, based on the proposed hypothesis, they should be expanded in the manuscript's conclusions. After all comments are addressed, the manuscript may be re-examined.

Author Response
|
Point-by-point response to Comments and Suggestions for Authors |
|
1 Review Comments 1: This manuscript certainly touches on an interesting topic. It is important to study the classic Bergmann's and Allen’s rules in animals in different geographic zones. However, the authors have devoted insufficient attention to describing their results. The manuscript cannot be published in its current form. The title of the manuscript should be amended. The text of the manuscript is jumbled in some chapters and should be moved to the appropriate chapters. In some methodological aspects of small mammal capture, the authors omit important information. This information should be added. The comparative section in the discussion needs to be expanded and additional literature on other geographic zones should be cited. The authors have obtained interesting results, presented them, and, based on the proposed hypothesis, they should be expanded in the manuscript's conclusions. After all comments are addressed, the manuscript may be re-examined. |
|
Response 1: Thank you for your insightful feedback on our study. We have thoroughly revised the manuscript to address your concerns, with major changes marked in red. 2 Specific comments |
|
Comments 1: Line 3 It is necessary to indicate where the study was conducted. |
|
Response 1: Lines 3-4 The title has been revised as “Community-level phenotypic adaptations of small mammals under rain-shadow dynamics in Baima Snow Mountain, Yunnan”. Comments 2: Line 22 Please indicate the number of species Response: Lines 23-24 The number of species is provided in the revision. Please write the research methods in detail. Response: Lines 25-29 The key research methods was added to this section. Line 36 standardized methods? Response: Lines 158-167 The standardized methods adhered to several protocols: 1) the distance between transects was consistent; 2) the same tools and baits were used; 3) all tools were placed at relatively fixed intervals; 4) the sampling efforts for all transects were similar. These methods facilitate replicated studies. These protocols were detailed in Line 38 laboratory? Response: The kidneys were processed, and the renal features were measured and calculated when back in laboratory. Comments 3: Lines 41-42 body size decreased with increasing temperature, aligning well with the Bergmann’s rule, This shouldn't be written here, as this rule is known for all animals. Only deviations from the rule should be written. Response 3: Lines 47-48 We described this result as “aligning well with conventional prediction”. Comments 4: Lines 100-106 Among which, renal structure, especially medullary thickness which directly points to the maximum length of the Henle loop , were frequently use as an estimate of urinary concentrating ability . This correlation suggests that animals inhabiting arid regions exhibit an increased proportion of renal medulla to optimize water utilization, a pattern that has been demonstrated at the community level and is reflected in the clustered functional structure of communities across both local and global scales .This is in Discussion? Response 4: We appreciate the reviewers 'feedback. As part of the theoretical foundation of this study, a brief introduction to this physiological characteristic is needed in the Introduction to enable us to formulate reasonable scientific hypotheses. Comments 5: Lines 107-110 Please clearly formulate the main aim of the study. Response 5: Lines 115-117 Thank you for prompting us to clarify the study's primary aim. We formulated our main aim of the study as: “Our research seeks to decipher the relative contributions of thermal, productivity, and humidity gradients in driving functional trait assembly of small mammal communities in the rain-shadow-impacted Baima Snow Mountains.”. Comments 6: Lines 139-140 This sentence should be rewritten in a strict scientific style.The region also boasts well-preserved natural habitats and rich wildlife, and it has been established as the protected Baimaxueshan National Nature Reserve. Response 6: Lines 147-150 Thank you for prompting greater scientific rigor. We have revised the sentence to: "This mountain also forms the core area of the Baimaxueshan National Nature Reserve, designated under China’s protected area system for in situ conservation of endemic flora and fauna characterized by high habitat integrity." Comments 7: Line 150 What kind of bait was used? Please write. Response 7: Lines 163-165 Thank you for prompting methodological clarification. We added ”The Sherman live traps were baited with sugar-free oatmeal, and the snaps were baited with fresh peanuts, while the buckets were deployed on the potential run path of small mammals without bait.” Comments 8: Line 151 How often were the traps checked? Response 8: L165-167 We added “The traps were deployed on the first day, checked and re-baited the following morning, and relocated to another transect/sub-transect on the third day.” Comments 9: Lines 152-153 Write a link to the source of literature. Response 9: Line 170 The reference have now been supplemented. Comments 10: Line 161 You already wrote this above. It needs to be combined.five external phenotypic features (BW, HB, TL, HF, and EL) Response 10: lines176-177 Based on your feedback, we have condensed this content. Comments 11: Line 216 All species must be given in the manuscript. Response 11: Lines 243-248 Thank you for your attention. We have listed all species. Comments 12: Lines 220-227;240-246There is very little description of the results. Response 12: Lines 254-260 ;276-282 We appreciate the reviewers' meticulous review and have provided a more detailed description of the results Comments 13: Line 227Temporal and spatial dynamics of the relative abundance of small mammal species allow us to assess many aspects of environmental variability (Andreychev 2021, Rui et al. 2025). Response 13: Line 327-329 The content has been added to the original text and references inserted. Comments 14: L344,345 for conservation governance,How?Please write Response 14: Lines 417-420 We articulate the rationale for identifying animal adaptive strategies to water dynamics for conservation purposes as follows: “given the rising frequency of extreme climate events—including severe droughts and intense rainfall—have led to increasing rate of extinctions, distributional and phenological changes, and species’ range shifts in wild animals in the current era ”. Comments 15: Line 483Add:Andreychev, A. Population structure and dynamics of small rodents and insectivorous mammals in a region of the Middle Volga, Russia. Biharean Biol. 2021, 15, 33–38.Rui, H.; Qin, S.; Jinfu, F.; Liqing, W.; Linbo, X.; Yuchuang, H.; Miaomiao, H.; Bobo, D.; Yanjun, T.; Yuheng, Zh.; Manduriwa. Temporal and Spatial Dynamics of Rodent Species Habitats in the Ordos Desert Steppe, China. Animals 2025, 15 (5), 721. Response 15: Lines 329 The above references have been inserted into the main text as references 60 and 61. |

Round 2
Reviewer 2 Report
Comments and Suggestions for Authors
Dear Authors,
I am satisfied with the revision of the manuscript. Corrections and additional data have been incorporated into the manuscript. The scientific and practical significance of the study of phenotypic adaptations of small mammals at the community level under rain-shadow conditions on Baima Snow Mountain, Yunnan, is beyond doubt. The review of the study results was selected appropriately, as were the statistical methods used for its analysis. The article has taken into account the comments on the methodology. The analysis and conclusion for each chapter are sufficient and unambiguous. The references have been adjusted. The results of previous studies by other authors have been taken into account. I recommend this work for the journal Animals.